# Modeling: Activity Concentration of Radon, Thoron, and Their Decay Products in Closed Systems

**DOI:** 10.3390/ijerph192416739

**Published:** 2022-12-13

**Authors:** Krystian Skubacz, Bogusław Michalik

**Affiliations:** Central Mining Institute, Silesian Centre for Environmental Radioactivity, Plac Gwarków 1, 40-166 Katowice, Poland

**Keywords:** radon and thoron progeny, activity concentration, potential alpha energy concentration, poorly ventilated spaces, calibration facility, model, application

## Abstract

The article presents a model for simulating changes in the activity concentration of radon and thoron as well as their progeny in closed or poorly ventilated systems. A system can be considered closed when a stream of radon and thoron flows into a space, but nothing comes out. It was also assumed that there may be devices or installations with a filtering system that would reduce the concentration of radon and thoron decay products. These assumptions may, therefore, correspond to a situation in which, in an isolated chamber, the calibration of radon hazard-monitoring devices is carried out, and nuclides are supplied from an emanation or flow through sources or well-isolated spaces in an environment where the source of nuclides is, for example, radon and thoron exhalation. The differential equations were formulated on the basis of the assumption that the activity concentration of radionuclides of concern in the space is uniform. The equations do not consider possible losses due to diffusion or the inertial or gravitational deposition of aerosols. If these phenomena have a limited impact on changes in the activity concentration of nuclides, the solutions provided may be used to simulate the activity concentration of radon and thoron and their decay products in any confined space assuming different boundary conditions.

## 1. Introduction

Radon (Rn-222) and thoron (Rn-220) isotopes belong to the uranium and thorium decay series, respectively. Their activity concentrations in the air are often enhanced in any type of confined space. Usually, together with their decay products (Figure 1), they make the largest contribution to the average annual dose of ionizing radiation that residents receive. For example, according to the US Environmental Protection Agency (EPA) and taking both artificial and natural sources of radiation into account, the share including radon and thoron reaches 37% of the 6.2 mSv annual effective dose per person [1]. Moreover, the agency concluded that radon exposure is the second leading cause of lung cancer after smoking. Recent epidemiological studies have shown that a statistically significant increase in the risk of lung cancer can be observed when the concentration is 100 Bq/m^3^, and it increases by 16% per 100 Bq/m^3^ increase during a prolonged exposure period [2]. For this reason, the Council Directive of the European Union established the basic safety standards for protection against the hazards related to ionizing radiation, which were passed in 2013; particular attention was paid to issues related to radon exposure [3].

Different solutions have aimed to reduce radon risk exist in many other countries as well, for example, Australia [4], Canada [5], Great Britain [6], the United States [7,8], and Poland [9]. The solutions used in Poland consist in sealing the sidewalls of mining excavations to reduce exhalation, limiting the outflow of radon from post-mining voids, and simply filtering the air from radon and thoron decay products. The model can also be used to predict the efficiency of such mitigation methods to some extent.

Moreover, according to a recent report published by the International Commission on Radiological Protection (ICRP), dose conversion factors have increased considerably [10], and now, the committed effective dose corresponding to the same radon and thoron progeny activity concentration is noticeably larger than before.

Generally, there are more efforts focused on exposure to radon than thoron. Sometimes, however, the opposite situation may be the case, e.g., in thoron-rich areas, such as the Loess Plateau within the provinces of Shanxi, Shaanxi, Gansu, and Henan (China). In the Gansu Province, more than 3 million people are used to living in cave dwellings. A typical cave dwelling there consists of one room with a single entrance and two windows in the front. It is made of local soil enriched with radionuclides from the thorium series. Therefore, the thoron activity concentration, 7–1085 Bq/m^3^, is considerably higher than that of radon, which equals 7–295 Bq/m^3^ [11].

Radon exposure is mainly evaluated through radon activity concentration measurements. However, radon’s short-lived decay products are responsible for more than 95% of the total dose. Therefore, to evaluate the dose on the basis of the radon gas activity concentration, specific equilibrium factors between radon and its decay products must be assumed. According to the ICRP report [10], the equilibrium factors for a reference worker (average breathing rate of 1.2 m^3^/h) are equal to 0.4 in indoor workplaces, 0.2 in mines, and 0.4 in tourist caves. As has been shown in different studies, the equilibrium factors can differ considerably from the values mentioned above [12,13,14], especially when mechanical ventilation is applied. According to a comprehensive study carried out by Chen and Harley [12], the equilibrium factors varied from 0.08 to 0.72 for active mines in 18 countries and from 0.10 to 0.85 for caves, tourist mines, and thermal spas. For this reason, it is much better to use active devices with a filtration system to directly measure the potential alpha energy concentration (PAEC), which can then be used to evaluate the appropriate dose directly. Such a solution is often recommended for underground mines with forced ventilation, where equilibrium factors can change significantly [7,8,15,16].

In the case of thoron, the issue of risk assessment is more complicated. Due to the mutual ratios of half-lives among this series of radionuclides, especially with thoron’s very short half-life, there is no equilibrium between thoron and its progeny, and it is almost impossible to consider the activity concentration of thoron as an indicator allowing even a rough estimation of the dose, as opposed to the situation with radon. Therefore, the ICRP report [10] includes conversion factors of the exposure for a reference worker to the committed effective dose of thoron, not just equilibrium factors. Because of this, the activity concentrations of thoron progeny or their potential alpha energy concentration must be measured directly to evaluate the committed effective dose.

Regardless of whether radon (thoron) or its progeny are measured, relevant devices used must be calibrated, which usually takes place in a chamber isolated from the environment where radon (thoron) and its progeny are present (Rn-222 (Rn): Po-218, Pb-214, Bi-214, and Po-214, and Rn-220 (Th): Po-216, Pb-212, Bi-212, Po-212, and Tl-208). The activity concentration of the radionuclides as well as radioactive equilibrium conditions in the chamber depend on its construction and the source of radon and thoron used.

The developed model was based on the analytical solutions of a system of differential equations describing decay sub-series starting with radon and thoron. Contrary to Bateman’s equations, the model provides for radon and thoron inflow, which may compensate for these decay series’ parent radionuclide decay. The model can be used to predict changes in the activity concentration of radon and thoron as well as all their decay products present inside poorly ventilated facilities, such as calibration chambers or working places in underground mines and caves, where radon or thoron flows in or is exhaled at a constant, known rate. Moreover, the presence of an additional device equipped with pumps and filtration systems working inside can be assumed, simulating either the calibration of active measuring equipment (in a chamber) or a risk mitigation method applied (in working areas) (Figure 2).

## 2. Materials and Methods

The model was developed on the basis of differential equations corresponding to the situation in which radon or thoron can flow into the system or be exhaled from an internal source at a constant rate. Moreover, inside, the system can operate devices equipped with filtering systems. After solving the equations, their correctness was tested using Mathematica software [17]. In order to facilitate the practical application of the model, Excel worksheets were created separately for radon and thoron, supported by the VBA (Visual Basic for Applications) language code [18], which enabled convenient calculations, as well as the automatic creation of charts for any parameters indicated, automatic changes to the chart ranges, their adaptation to the user’s needs, and control of the correctness of the values entered (Figure 3). In addition to the activity concentration of radionuclides, the potential alpha energy concentration and the partial contribution of individual radionuclides to this value in absolute and percentage terms were also assessed.

The application named “Application.xlsm” performs all the tasks specified by the model and can be downloaded together with the publication (Appendix A).

## 3. Results

The model assumes that the radon or thoron is delivered with a certain constant efficiency E_1_ (Bq/s) to the system with volume V (m^3^), and at the same time, the parent nuclides and their decay products undergo radioactive decay. Additionally, the activity concentration of the decay products can be reduced owing to the operation of installations or devices equipped with filtration systems of filtration efficiency F, through which air flows with a flow rate of Q_D_ (m^3^/s). Taking such assumptions into consideration, the differential equations for activity concentration C_i_ (Bq/m^3^) of the i-radionuclide must comply with the following rules:(1)$$\frac{\partial C_{1}}{\partial t}=S_{1}-\lambda_{1}C_{1}\;\;\;\;\;\;\;\;\;\;\;\;\frac{\partial C_{i}}{\partial t}=-\left(R_{D}+\lambda_{i}\right)C_{i}+\varepsilon_{i}\lambda_{i}C_{i-1}\;\;\;\;\;\;i\;>1$$
where the S_1_ and R_D_ parameters are equal:S_1_ = E_1_/V (Bq × m^−3^ × s^−1^)        R_D_ = FQ_D_/V (s^−1^)(2)
and ε_i_ are the branch ratios.

Solutions to these equations are given below (3)–(7):(3)$$C_{1}=C_{o1}e^{-\lambda_{1}t}+S_{1}\left(\frac{1}{\lambda_{1}}-\frac{1}{\lambda_{1}}e^{-\lambda_{1}t}\right)$$
(4)$$C_{2}=\varepsilon_{2}C_{o1}\left(\frac{\lambda_{2}}{\lambda_{21R}}e^{-\lambda_{1}t}-\frac{\lambda_{2}}{\lambda_{21R}}e^{-\lambda_{2R}t}\right)+C_{o2}e^{-\lambda_{2R}t}+\;{\;\varepsilon }_{2}S_{1}\left(\frac{\lambda_{2}}{\lambda_{1}\lambda_{2R}}-\frac{\lambda_{2}}{\lambda_{1}\lambda_{21R}}e^{-\lambda_{1}t}+\frac{\lambda_{2}}{\lambda_{2R}\lambda_{21R}}e^{-\lambda_{2R}t}\right)$$
(5)$$C_{3}=\varepsilon_{2}\varepsilon_{3}C_{o1}\left(\frac{\lambda_{2}\lambda_{3}}{\lambda_{21R}\lambda_{31R}}e^{-\lambda_{1}t}-\frac{\lambda_{2}\lambda_{3}}{\lambda_{21R}\lambda_{32}}e^{-\lambda_{2R}t}-\frac{\lambda_{2}\lambda_{3}}{\lambda_{31R}\lambda_{23}}e^{-\lambda_{3R}t}\right)+\;{\;\varepsilon }_{3}C_{o2}\left(\frac{\lambda_{3}}{\lambda_{32}}e^{-\lambda_{2R}t}+\frac{\lambda_{3}}{\lambda_{23}}e^{-\lambda_{3R}t}\right)+C_{o3}e^{-\lambda_{3R}t}+\;\varepsilon_{2}\varepsilon_{3}S_{1}\left(\begin{array}{c} \frac{\lambda_{2}\lambda_{3}}{\lambda_{1}\lambda_{2R}\lambda_{3R}}-\frac{\lambda_{2}\lambda_{3}}{\lambda_{1}\lambda_{21R}\lambda_{31R}}e^{-\lambda_{1}t}+\frac{\lambda_{2}\lambda_{3}}{\lambda_{2R}\lambda_{21R}\lambda_{32}}e^{-\lambda_{2R}t}+\frac{\lambda_{2}\lambda_{3}}{\lambda_{3R}\lambda_{31R}\lambda_{23}}e^{-\lambda_{3R}t} \end{array}\right)$$
(6)$$C_{4}=\varepsilon_{2}\varepsilon_{3}\varepsilon_{4}C_{o1}\left(\begin{array}{c} \frac{\lambda_{2}\lambda_{3}\lambda_{4}}{\lambda_{21R}\lambda_{31R}\lambda_{41R}}e^{-\lambda_{1}t}-\frac{\lambda_{2}\lambda_{3}\lambda_{4}}{\lambda_{21R}\lambda_{32}\lambda_{42}}e^{-\lambda_{2R}t}-\\ \frac{\lambda_{2}\lambda_{3}\lambda_{4}}{\lambda_{31R}\lambda_{23}\lambda_{43}}e^{-\lambda_{3R}t}-\frac{\lambda_{2}\lambda_{3}\lambda_{4}}{\lambda_{41R}\lambda_{24}\lambda_{34}}e^{-\lambda_{4R}t} \end{array}\right)+\;{\;\varepsilon }_{3}\varepsilon_{4}C_{o2}\left(\begin{array}{c} \frac{\lambda_{3}\lambda_{4}}{\lambda_{32}\lambda_{42}}e^{-\lambda_{2R}t}+\frac{\lambda_{3}\lambda_{4}}{\lambda_{23}\lambda_{43}}e^{-\lambda_{3R}t}+\frac{\lambda_{3}\lambda_{4}}{\lambda_{24}\lambda_{34}}e^{-\lambda_{4R}t} \end{array}\right)+\;{\;\varepsilon }_{4}C_{o3}\left(\frac{\lambda_{4}}{\lambda_{43}}e^{-\lambda_{3R}t}+\frac{\lambda_{4}}{\lambda_{34}}e^{-\lambda_{4R}t}\right)+C_{o4}e^{-\lambda_{4R}t}+\;\varepsilon_{2}\varepsilon_{3}\varepsilon_{4}R\left(\begin{array}{c} \frac{\lambda_{2}\lambda_{3}\lambda_{4}}{\lambda_{1}\lambda_{2R}\lambda_{3R}\lambda_{4R}}-\frac{\lambda_{2}\lambda_{3}\lambda_{4}}{\lambda_{1}\lambda_{21R}\lambda_{31R}\lambda_{41R}}e^{-\lambda_{1}t}+\frac{\lambda_{2}\lambda_{3}\lambda_{4}}{\lambda_{2R}\lambda_{21R}\lambda_{32}\lambda_{42}}e^{-\lambda_{2R}t}+\\ \frac{\lambda_{2}\lambda_{3}\lambda_{4}}{\lambda_{3R}\lambda_{31R}\lambda_{23}\lambda_{43}}e^{-\lambda_{3R}t}+\frac{\lambda_{2}\lambda_{3}\lambda_{4}}{\lambda_{4R}\lambda_{41R}\lambda_{24}\lambda_{34}}e^{-\lambda_{4R}t} \end{array}\right)$$
(7)$$\begin{array}{cc} \; & C_{5}=\varepsilon_{2}\varepsilon_{3}\varepsilon_{4}\varepsilon_{5}C_{o1}\left(\begin{array}{c} \frac{\lambda_{2}\lambda_{3}\lambda_{4}\lambda_{5}}{\lambda_{21R}\lambda_{31R}\lambda_{41R}\lambda_{51R}}e^{-\lambda_{1}t}-\frac{\lambda_{2}\lambda_{3}\lambda_{4}\lambda_{5}}{\lambda_{21R}\lambda_{32}\lambda_{42}\lambda_{52}}e^{-\lambda_{2R}t}-\\ \frac{\lambda_{2}\lambda_{3}\lambda_{4}\lambda_{5}}{\lambda_{31R}\lambda_{23}\lambda_{43}\lambda_{53}}e^{-\lambda_{3R}t}-\frac{\lambda_{2}\lambda_{3}\lambda_{4}\lambda_{5}}{\lambda_{41R}\lambda_{24}\lambda_{34}\lambda_{54}}e^{-\lambda_{4R}t}-\\ \frac{\lambda_{2}\lambda_{3}\lambda_{4}\lambda_{5}}{\lambda_{51R}\lambda_{25}\lambda_{35}\lambda_{45}}e^{-\lambda_{5R}t} \end{array}\right)+\\ \; & \varepsilon_{3}\varepsilon_{4}\varepsilon_{5}C_{o2}\left(\begin{array}{c} \frac{\lambda_{3}\lambda_{4}\lambda_{5}}{\lambda_{32}\lambda_{42}\lambda_{52}}e^{-\lambda_{2R}t}+\frac{\lambda_{3}\lambda_{4}\lambda_{5}}{\lambda_{23}\lambda_{43}\lambda_{53}}e^{-\lambda_{3R}t}+\\ \frac{\lambda_{3}\lambda_{4}\lambda_{5}}{\lambda_{24}\lambda_{34}\lambda_{54}}e^{-\lambda_{4R}t}+\frac{\lambda_{3}\lambda_{4}\lambda_{5}}{\lambda_{25}\lambda_{35}\lambda_{45}}e^{-\lambda_{5R}t} \end{array}\right)+\\ \; & \varepsilon_{4}\varepsilon_{5}C_{o3}\left(\frac{\lambda_{4}\lambda_{5}}{\lambda_{43}\lambda_{53}}e^{-\lambda_{3R^{t}}}+\frac{\lambda_{4}\lambda_{5}}{\lambda_{34}\lambda_{54}}e^{-\lambda_{4R^{t}}}+\frac{\lambda_{4}\lambda_{5}}{\lambda_{35}\lambda_{45}}e^{-\lambda_{5}R}\right)+\\ \; & \varepsilon_{5}C_{o4}\left(\frac{\lambda_{5}}{\lambda_{54}}e^{-\lambda_{4R}t}+\frac{\lambda_{5}}{\lambda_{45}}e^{-\lambda_{5R}t}\right)+C_{o5}e^{-\lambda_{5R}t}+\\ \; & \varepsilon_{2}\varepsilon_{3}\varepsilon_{4}\varepsilon_{5}S_{1}\left(\begin{array}{c} \frac{\lambda_{2}\lambda_{3}\lambda_{4}\lambda_{5}}{\lambda_{1}\lambda_{2R}\lambda_{3R}\lambda_{4R}\lambda_{5R}}-\frac{\lambda_{2}\lambda_{3}\lambda_{4}\lambda_{5}}{\lambda_{1}\lambda_{21R}\lambda_{31R}\lambda_{41R}\lambda_{51R}}e^{-\lambda_{1}t}+\\ \frac{\lambda_{2}\lambda_{3}\lambda_{4}\lambda_{5}}{\lambda_{2R}\lambda_{21R}\lambda_{32}\lambda_{42}\lambda_{52}}e^{-\lambda_{2R}t}+\frac{\lambda_{2}\lambda_{3}\lambda_{4}\lambda_{5}}{\lambda_{3R}\lambda_{31R}\lambda_{23}\lambda_{43}\lambda_{53}}e^{-\lambda_{3R}t}+\\ \lambda_{2}\lambda_{4R}\lambda_{41R}\lambda_{24}\lambda_{34}\lambda_{54}e^{-\lambda_{4R}t}+\frac{\lambda_{2}\lambda_{3}\lambda_{4}\lambda_{5}}{\lambda_{5R}\lambda_{51R}\lambda_{25}\lambda_{35}\lambda_{45}}e^{-\lambda_{5R}t} \end{array}\right)\\ \; \end{array}$$

To simplify the notation of quite complex functions, the following conventions were introduced: λ_ij_ ≡ λ_i_ − λ_j_, λ_iR_ ≡ λ_i_ + R_D_ and λ_ijR_ ≡ λ_i_ − λ_j_ + R_D_, where the quantities λ_i_ or λ_j_ are the decay constants of individual radionuclides, and indices i, j =1, 2, 3, 4, and 5 refer respectively to Rn-222, Po-218, Pb-214, Bi-214 and Po-214 or Rn-220, Po-216, Pb-212, and Bi-212. For radionuclides Po-212 and Tl-208, the corresponding index is i, j = 5. It can be assumed that the branch ratios ε_i_ are equal to one, except for the isotopes Po-212 and Tl-212, for which ε_5_ = 0.638 (Po-212) or ε_5_ = 0.362 (Tl-208).

There was no possibility to use these solutions and present parameters, such as time, filtration, etc., in the form of symbolic analytic functions of other parameters. This is why we enclosed a flexible tool to solve all the problems covered by these equations. The tool can be used to achieve not only the charts but also the values related to activity concentration, potential alpha energy concentration, and equilibrium coefficients for the given time. These results can be obtained for up to 1000 sub-intervals, into which any interval indicated by the user can be divided and are available in the hidden cells available in the supporting material using the standard MS Excel option “display cells value”.

Equations (3)–(7) also describe situations where radon or thoron of known activity is introduced into the space at once so that their activity concentration is initially C_o1_, and there is no longer any inflow. The differential equations describing this situation are as follows:(8)$$\frac{\partial C_{1}}{\partial t}=-\lambda_{1}C_{1}\;\;\;\;\;\;\;\;\;\frac{\partial C_{i}}{\partial t}=-\lambda_{i}C_{i}+\varepsilon_{i}\lambda_{i}C_{i-1}\;\;\;\;\;\;i>1$$
and their solutions are classic Bateman functions [19] representing the secular equilibrium state and no equilibrium due to decay law in the cases of radon and thoron and their progeny, respectively. Solutions of differential Equation (8) could be obtained after transforming Formulas (3)–(7) by the substitution of S_1_ = 0 and R_D_ = 0.

Assuming, in turn, that the activity concentration of radon or thoron in the chamber remains constant, i.e., radon or thoron inflow compensates for the radioactive decay completely, the set of differential equations is as follows:(9)$$\frac{\partial C_{1}}{\partial t}=0\;\;\;\;\;\;\;\;\;\;\;\;\frac{\partial C_{i}}{\partial t}=-\lambda_{i}C_{i}+\varepsilon_{i}\lambda_{i}C_{i-1}\;\;\;\;\;\;i>1$$
and their solutions are obtained after transforming Formulas (3)–(7) by first substituting S_1_ = 0 and R_D_ = 0 and then λ_1_ = 0. In such a case, under real conditions, an equilibrium state can be reached between thoron and its decay product regardless of decay law.

When analyzing the dependencies (3)–(7), it was possible to notice that if the initial activity concentration was C_o1_ = S_1_/λ_1_, then the constant inflow of radon and thoron to the chamber compensated for its radioactive decay, and as a result, its concentration did not change. This state could also be reached after a certain time has elapsed (Figure 4), which, however, was approximately 40 days for radon and approximately 10 min for thoron, irrespective of the radon or thoron inflow rate or the space volume. The concentration of radon or thoron, S_1_/λ_1_, was also the concentration that the activity concentration of these isotopes in the chamber approached asymptotically. Figure 4 also illustrates the effect of the operation of the installation with a filtration system inside the space. If the R_D_ parameter totaled zero, the radon decay products would have been in secular equilibrium with radon after about 3–4 h; however, their constant loss made it impossible to achieve this state.

Depending on the activity concentration, some denominators include elements such as λi − λ_1_ + R_D_ (i ≠ 1), which under certain conditions may turn into singularities in these equations, becoming equal to zero. In the case of radon and its progeny, this was not possible because the differences λ_i_ − λ_1_ are always greater than zero. However, in the case of thoron, such situations may take place if R_D_ = λ_1_ − λ_3_, R_D_ = λ_1_ − λ_4_, or R_D_ = λ_1_ − λ_5_ (for Tl-208). The contribution to the activity concentration of each isotope is made up of its own initial activity concentration, the initial activity concentrations of its parents, and the inflow of radon or thoron (S_1_ parameter). Apart from the inflow itself, this contribution decreases over time. A drop in the activity concentrations of radon and thoron progeny could be accelerated by the operation of active devices inside the space or chamber (R_D_ parameter). If the R_D_ parameter reached a high value, their activity concentration reduction was not compensated for by other factors, and the chain was broken. In practice, reaching the critical conditions is unlikely for a calibration chamber, because, for example, for a space with a volume of 20 m^3^, the flow rates in the operating measuring devices with a filtering efficiency of 100% would have to be approximately 15,000 L/min (C_3_ and C_4_) and approximately 10,000 L/min (C_5_ for Tl-208), and they would have to be approximately 750 L/min and about 500 L/min, respectively, for a 1 m^3^ chamber volume.

Figure 5 and Figure 6 show the simulation results of a situation in which radon was supplied to the chambers with a volume of 20 m^3^ and 1 m^3^, respectively, at a rate of 1 Bq/s. The device, through which air flowed at a rate of 10 L/min with a filtration efficiency of 100%, operated in the system from 4 to 6 h simultaneously. For larger systems, this had no major impact on the activity concentrations, and only a minor effect of the operating device was noticeable in the graph (Figure 5). However, it was much more pronounced in the 1 m^3^ system, where the effects of its operation disappeared after only about two hours after switching off the device (Figure 6).

The situation was different in the case of thoron. The constant delivery of thoron into a system in which no filtering devices were in operation, led to equilibrium after approximately 70 h (Figure 7). Then, the concentrations of all decay products reached the highest possible values. The volume of the chamber affected only the absolute values of the activity concentrations, but it did not affect the ratio of decay product activity concentration or the length of the process. Initially, for about 10 min, almost 100% of the contribution to the potential alpha energy concentration (PAEC) corresponded to Pb 212. Over time, it decreased steadily, but only to approximately 90% (Figure 8). The remaining 10% was related to Bi 212, which means that detecting these isotopes is crucial for the assessment of the potential alpha energy concentration, and the other isotopes can be ignored.

Changes in activity concentrations in the case of thoron delivery to the system with operating devices inside are much more significant than for radon in a comparable situation (Figure 9 and Figure 10). They already became clearly visible in a large chamber with a volume of 20 m^3^. For obvious reasons, however, the situation for smaller volumes was much more violent. As a consequence, a sharp change in the potential alpha energy concentration was observed (Figure 11 and Figure 12). Using smaller chambers was more beneficial, as higher potential alpha energy concentrations could be obtained with the same thoron delivery rate. Comparing the achievable potential alpha energy concentrations for radon and thoron inflowing with the same efficiency, its value inside the radon-filled chamber will always be significantly higher.

No aerosol removal mechanisms, such as diffusion, inertia, or gravity, were taken into account in the model. These depend on the aerosol size distribution, how the radionuclides will be bound to environmental aerosols, convection movements inside the chamber (caused for example by mixing air), chamber design, and environmental conditions (temperature and pressure). These conditions can be controlled, to some extent, by creating an appropriate atmosphere inside the chamber. Gravity settlement is important for larger aerosols. The terminal settling velocity of gravitational deposition for objects with a size of 100 nm and density of 1g/cm^3^ is in the order of 3 × 10^−7^ m/s [20]. Since this was achieved in far less than 1 ms, it can be assumed that this is the speed at which such objects fall to the bottom, a rather low value considering the time scale of the experiments. However, for objects with a size of 1 µm, the terminal settling velocity is two orders of magnitude higher, meaning that in standing air and under normal conditions (temperature and pressure), they would fall to the bottom of the chamber from a height of 1 m after about 9 h. Reducing the size of an object reduces gravitational deposition. At the same time, the significance of inertia in the aerosol removal process also decreases. On the other hand, the diffusion coefficient and the losses due to the plate-out effect increase, particularly in the case of strong convection currents inside a space, for example, as a result of its ventilation.

## 4. Conclusions

The model presented herein can be a useful tool for simulating the activity concentration of radionuclides from radon and thoron decay sub-series in the air inside confined or poorly ventilated spaces. Assuming facilities with a specific volume and a radon/thoron source with a specific capacity and characteristic (continuous or instantaneous), the model lets one follow each radionuclide’s activity concentration for the suite of radionuclides of concern. Moreover, assuming a device with filtering systems inside a facility, the model indicates the extent to which its operation can affect the existing conditions.

Such features let one use this model for a radon/thoron measuring device calibration facility development and calibration process planning. In cases of devices equipped with filtering systems, maintaining stable conditions is easier to achieve in larger facilities. On the other hand, for large chambers, an appropriately efficient source must be used to achieve sufficiently high activity concentrations, in particular in the case of thoron. As for calibration, a suitable method for measuring the reference potential alpha energy concentration must be applied [21]; such an application of the model allows also its validation and the quantification of the effects of parameters not included in the model yet.

Apart from any calibration processes and facilities, this model can be applied to well-isolated spaces in environments where natural radionuclides are present and lead to radon and thoron exhalation, for example, workplaces, especially underground ones. In such cases, a device with a filtering system put inside may be considered as a solution applied to mitigate the radiation hazard caused by radon as well as thoron, and the model could simulate the mitigation temporal efficiency.

Last but not least is this model’s educational potential. Charting radon isotopes and their decay product activity concentration under different conditions lets one understand decay rules and radionuclide equilibria and explains why one radionuclide can contribute more to the dose than the other.

## Figures and Tables

**Figure 1 ijerph-19-16739-f001:**
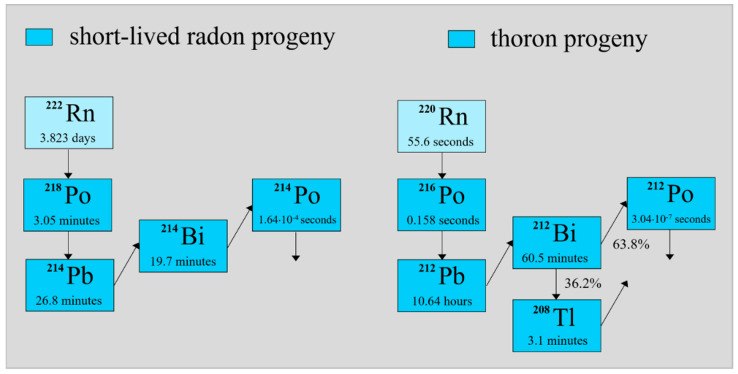
Radon and thoron decay products and their half-lives.

**Figure 2 ijerph-19-16739-f002:**
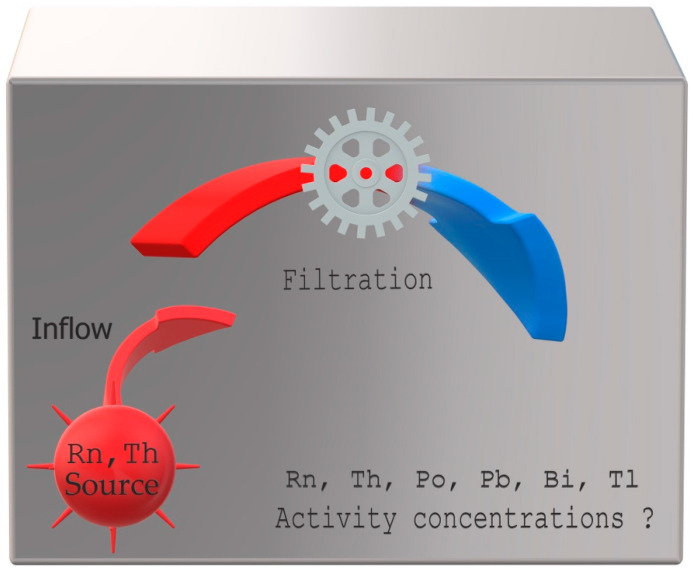
Basic assumptions of the model.

**Figure 3 ijerph-19-16739-f003:**
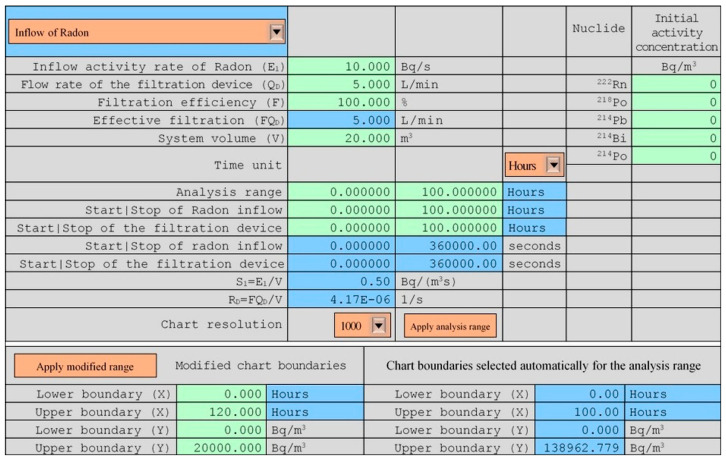
User interface. The meaning of the colors is as follows: the orange objects are the forms controls of the excel developer, the blue cells contain the rules, and should be not changed, and the green cells are designated for a user, who can introduce there the needed values.

**Figure 4 ijerph-19-16739-f004:**
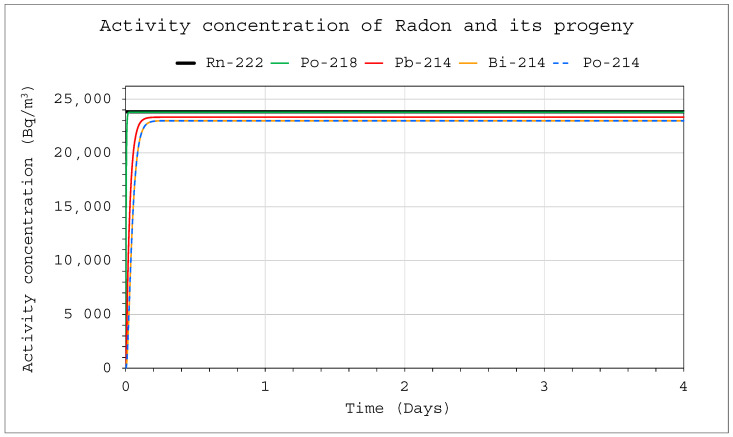
Activity concentration of radon and its progeny inside the system for the following conditions: E_1_ = 1 Bq/s (continuously), Q_D_ = 10 L/min (continuously), F = 100%, V = 20 m^3^, C_o1_ = S_1_/λ_1_, Coi = 0 for i ≠ 1.

**Figure 5 ijerph-19-16739-f005:**
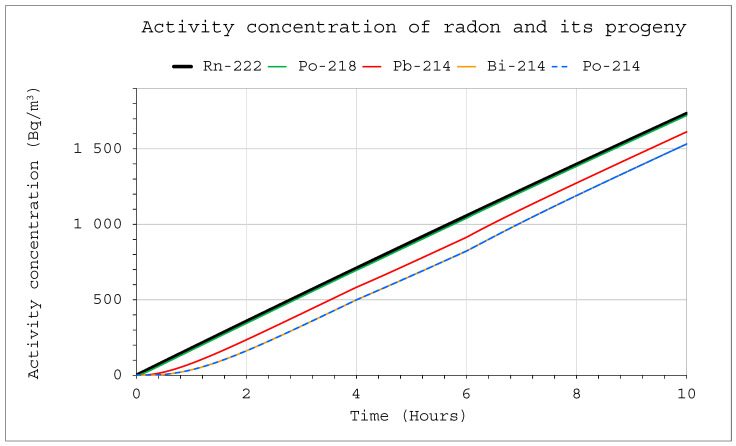
Activity concentration inside the system for the following conditions: E1 = 1 Bq/s (continuously), V = 20 m^3^, C_oi_ = 0. The filtration was occurring in the range of 4–6 h with a flow rate of Q_D_ = 10 L/min and a filtration efficiency of F = 100%.

**Figure 6 ijerph-19-16739-f006:**
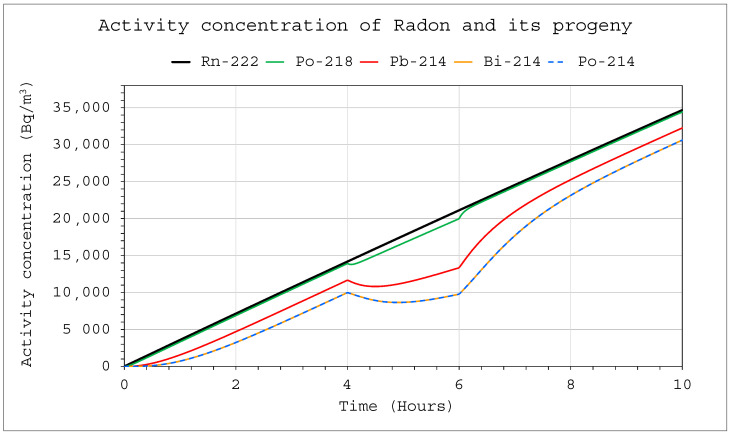
Activity concentration inside the system for the following conditions: E_1_ = 1 Bq/s (continuously), V = 1 m^3^, C_oi_ = 0. The device was operating in the range of 4–6 h with a flow rate of Q_D_ = 10 L/min and a filtration efficiency of F = 100%.

**Figure 7 ijerph-19-16739-f007:**
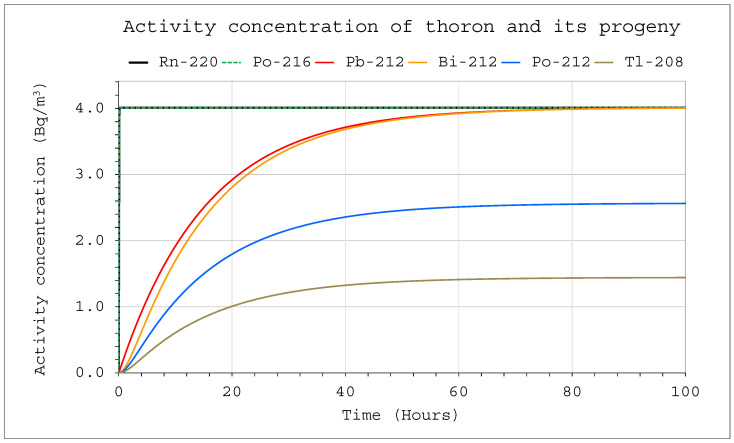
Activity concentration inside the system for the following conditions: inflow of thoron throughout the analyzed period at a rate of E_1_ = 1 Bq/s, V = 20 m^3^, C_oi_ = 0. No filtration system was operating inside.

**Figure 8 ijerph-19-16739-f008:**
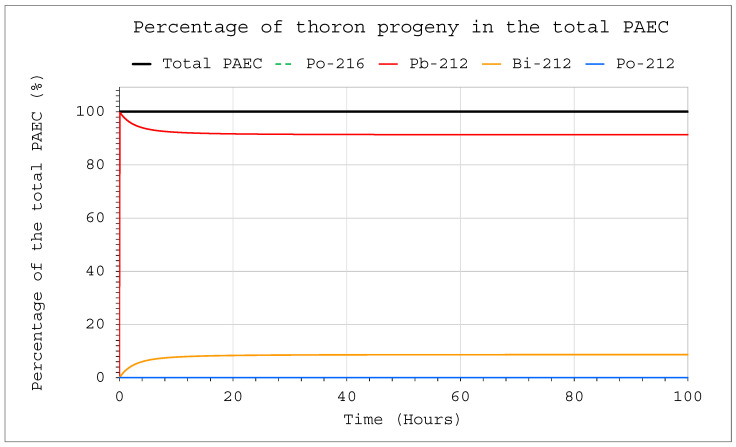
Percentage of the individual thoron progeny in the total PAEC inside the system for the following conditions: inflow of thoron throughout the analyzed period at a rate of E_1_ = 1 Bq/s, V = 20 m^3^, C_oi_ = 0. No operating filtration system was operating inside.

**Figure 9 ijerph-19-16739-f009:**
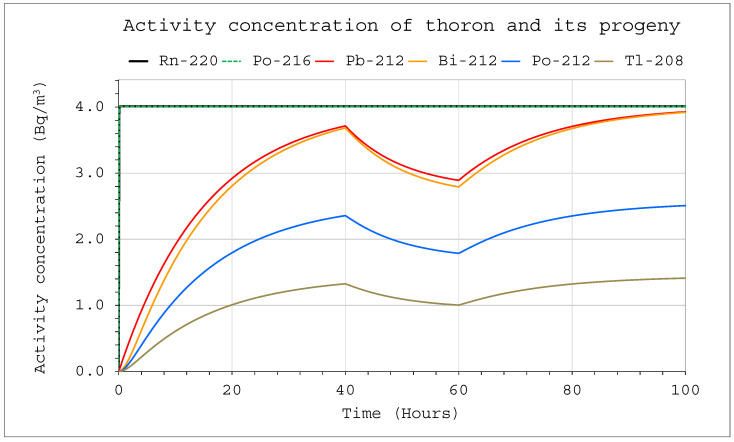
Activity concentration of thoron and its progeny inside the system for the following conditions: inflow of thoron throughout the analyzed period at a rate of E_1_ = 1 Bq/s, V = 20 m^3^, C_oi_ = 0. The filtration system was operating in the range of 40–60 h with a flow rate of Q_D_ = 10 L/min and a filtration efficiency of F = 100%.

**Figure 10 ijerph-19-16739-f010:**
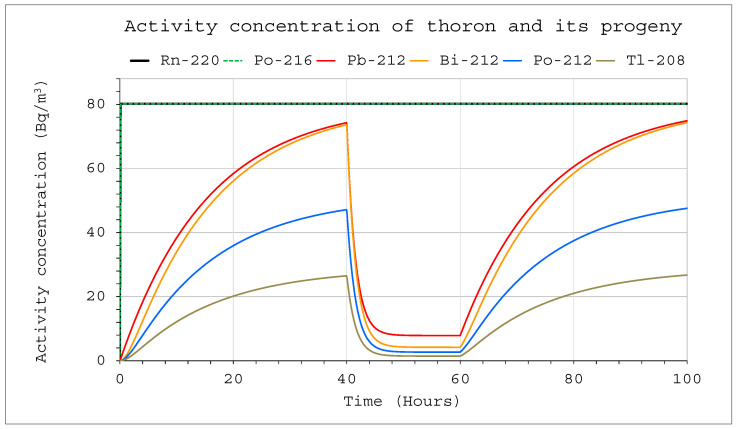
Activity concentration of thoron and its progeny inside the system for the following conditions: inflow of thoron throughout the analyzed period at a rate of E_1_ = 1 Bq/s, V = 1 m^3^, C_oi_ = 0. The filtration system was operating in the range of 40–60 h with a flow rate of Q_D_ = 10 L/min and a filtration efficiency of F = 100%.

**Figure 11 ijerph-19-16739-f011:**
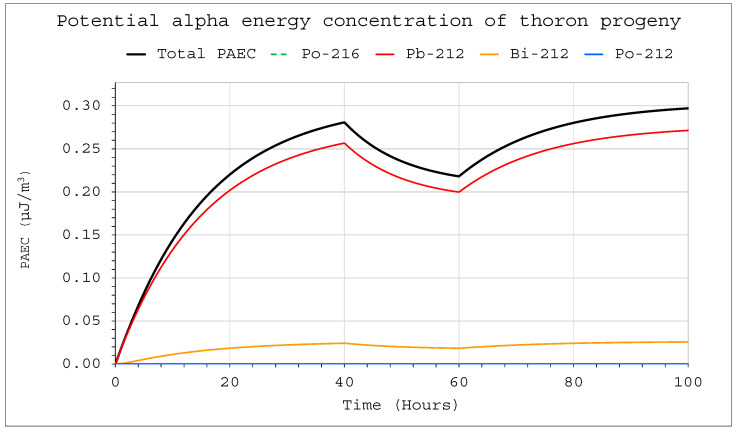
Potential alpha energy concentration of thoron progeny inside the system for the following conditions: inflow of thoron throughout the analyzed period at a rate of E_1_ = 1 Bq/s, V = 20 m^3^, C_oi_ = 0. The filtration system was operating in the range of 40–60 h with a flow rate of Q_D_ = 10 L/min and a filtration efficiency of F = 100%.

**Figure 12 ijerph-19-16739-f012:**
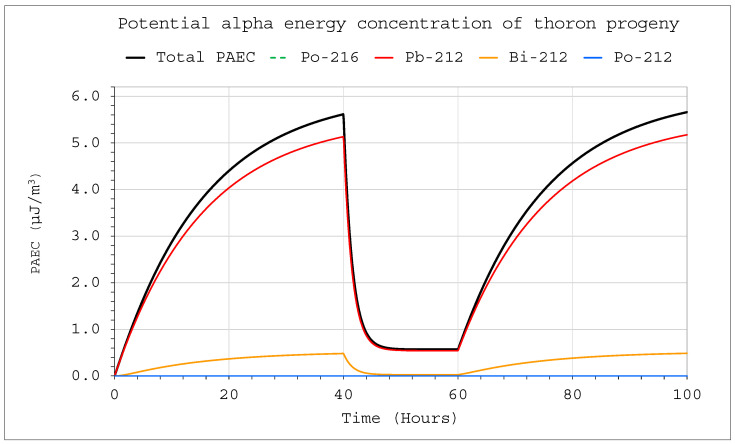
Potential alpha energy concentration of thoron progeny inside the system for the following conditions: inflow of thoron throughout the analyzed period at a rate of E_1_ = 1 Bq/s, V = 1 m^3^, C_oi_ = 0. The filtration system was operating in a range of 40–60 h with a flow rate of Q_D_ = 10 L/min and filtration efficiency of F = 100%.

## Data Availability

All the data related to the described model are available in the Appendix A as a part of the excel file.

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
