# Peer review of "Modeling: Activity Concentration of Radon, Thoron, and Their Decay Products in Closed Systems"

_ijerph, 2022, doi:10.3390/ijerph192416739_

Round 1
Reviewer 1 Report
An Excel-based computer code is presented which allows determining the concentration of radon (Rn-222, and Rn-220) and its progeny in a closed system under different boundary conditions. The solution of the system of differential equations is well known since the beginning of the knowledge of the natural decay series. Nevertheless, the Excel-based computer code seems to be a very convenient tool to calculate the radon/progeny concentration in different applications. The code is tailored for radon calibration chambers but I can see a good potential for such a code in an expansion for a dynamic behavior of the progeny, especially taking into account additional plate-out effects etc. Maybe this can be used to get to a better understanding of the equilibrium factor and its behavior under different conditions. Especially for thoron there may be a concentration gradient e.g. to a wall source and therefore an increased plate-out to the wall will occur, depending on the air-movement in the investigated room.
Although, there is no essential new scientific finding, I support the publication of this paper in connection with the Excel-based code because it is a really beautiful tool for calculating Rn/progeny concentrations as a function of time under adjustable conditions.
Some additional comments:
Is fig. 2 necessary?
Line 128: epsilon (branching ratio) should be explained at the first occurrance.
Line 303: ...[20]. Such... ..allows...
Line 307: For thoron too?
Author Response
Thanks a lot for your revision. Our answers are included in the pdf file.

Reviewer 2 Report
The authors discussed a mathematical model for simulating changes in the activity concentration of radon and thoron in closed or poorly ventilated systems. They assumed a closed system in which a stream of radon and thoron flows into the system with no outflow from the system. The model takes form as a system of differential equations which does not consider possible losses due to diffusion and inertial or gravitational deposition of aerosols. They used the model to simulate the activity concentration of radon and thoron and their decay assuming different boundary conditions and the installations of a filtering system, which will reduce the concentration of radon and thoron decay products. They able to predict the effect of this installation of the filtering system for a given filtering time periods. It is an interesting manuscript, however, there are some issues that should be addressed by the authors:
1. Introduction line 39: Please give some examples of the solution to reduce the radon risk after the sentence “Different solutions aimed at reducing radon risk exist in many other countries”.Please also explain what is the connection to the model presented in the manuscript.
2. There is no citation in the text to Figures 1 and 3. Please add.
3. Line 172: From Figure 4, the equilibrium solutions of the system (3)-(7) approches the equilibrium solution of the system. It would be better if the authors show the mathematical expression of these equilibria, by solving the steady state equations in (3)-(7).
4. Line 174: It is also claimed that the time needed to arrive to the equilibria are 40 days and 10 minutes. How to find the exact values of these numbers analytically, not only visually from Figure 4.
5. Line 212: Figure 7 shows the solution when there is no filtering devices. It leads to equilibrium after approx 70 hours. Figure 9 shows the solution when there is a filtering devices in the time interval 40-60 hours (total filtering time 20 hours). The results in Figure 9 shows that the solution approaches the equilibrium solution in more than 100 hours. It is nice if the authors could give the exact values of the time needed to roughly arrive to the equilibrium analytically and make prediction for any length of filtering time and any filtering efficiency. The time needed could be presented as a function of filtering time and filtering efficiency. Further tables or graphs could also produced. This would show the significance of their works in the field and makes the manuscript stronger.
Author Response

(The authors gave the same response as above.)

Round 2
Reviewer 2 Report
Dear Authors,
Thank you for submiting the revised version of the manuscript.